# Trend and determinants of minimum dietary diversity consumption change among children aged 6–23 months in Ethiopia from 2011 to 2019: A multivariate decomposition analysis

Abel Endawkie[1]*, Lakew Asmare[1], Hiwot Tadesse Alemu[2], Demiss Mulatu Geberu[2], Asebe Hagos[2], Melak Jejaw[2], Kaleab Mesfin Abera[3], Misganawu Guadie Tiruneh[2], Kaleb Assegid Demissie[2], Yawkal Tsega[4], Adina Yeshambel Belay[2], Wubshet Debebe Negash[2], Amare Mesfin Workie[5], Lamrot Yohannes[6], Mihret Getnet[7,8], Nigusu Worku[2]

1 Department of Epidemiology and Biostatistics, School of Public Health, College of Medicine and Health Sciences, Wollo University, Dessie, Ethiopia, 2 Department of Health Systems and Policy, Institute of Public Health, College of Medicine and Health Science, University of Gondar, Gondar, Ethiopia, 3 Department of Health Systems and Policy, Institute of Public Health, College of Medicine and Health Science, Wollo University, Dessie, Ethiopia, 4 Department of Health Systems Management and Policy, School of Public Health, College of Medicine and Health Science, Wollo University, Dessie, Ethiopia, 5 Department of Nutrition, Institute of Public Health, College of Medicine and Health Science, University of Gondar, Gondar, Ethiopia, 6 Department of Environmental and Occupational Health and Safety, Institute of Public Health, College of Medicine and Health Science, University of Gondar, Gondar, Ethiopia, 7 Department of Human Physiology, School of Medicine, College of Medicine and Health Science, University of Gondar, Gondar, Ethiopia, 8 Department of Epidemiology and Biostatistics, Institute of Public Health, College of Medicine and Health Science, University of Gondar, Gondar, Ethiopia

* abelendawkie@gmail.com

**Data Availability Statement:** The data was obtained from The DHS Program (https://

## Abstract

### Background

Sustainable Development Goals 2 target 2.2 aimed to **end all forms of malnutrition by 2030**. However, the prevalence of inadequate minimum dietary diversity (MDD) is high in Ethiopia and the evidence of trends and determinants of MDD consumption change among children over time is limited. Therefore, this study aimed to determine trends and determinants of MDD consumption change among children aged 6 to 23 months in Ethiopia from 2011–2019 Demographic and Health Survey (DHS) data.

### Method

The study used the data on nationally representative weighted samples of 6,971, 7,467, and 3,154 children aged 6–23 months from the 2011, 2016, and 2019 rounds of the DHS. Trend and Multivariate Decomposition analysis was conducted to determine MDD consumption change from 2011–2016, and 2016–2019 into two components: one that was explained by differences in the level of the determinants (endowment), and the other component was explained by differences in the effect of the covariates on the outcome (coefficient effects).

dhsprogram.com/) by requesting from The DHS Program after creating an account and submitting a concept note with project title. More access information can be found on The DHS Program website(https://dhsprogram.com/data/Access-Instructions.cfm). We confirmed that interested researchers would be able to access these data in the same manner as the authors. We also confirm that we had no special access privileges that others would not have.

**Funding:** The author(s) received no specific funding for this work.

**Competing interests:** The authors have declared that no competing interests exist.

**Abbreviations:** CSA, Central Staticall Agency; EA, Enumeration Area; MDD, Minimum Dietary Diversity; DHS, Demographic and Health Survey; EDHS, Ethiopian Demographic and Health Survey, mvdcmp: Multivariate decomposition; SDG, Sustainable Development Goals; SSA, Sub-Saharan Africa; WHO, World Health Organization.

## Result

The trends analysis showed that adequate MDD consumption significantly increased from 2% to 10.41% from 2011 to 2016 but decreased from 10.41% to 7.11% from 2016 to 2019 in Ethiopia. The compositional and behavioral change factors like maternal age, occupational status of parents, sex of the household head, wealth index, residence, and sex of child statistically contributed to changes in MDD consumption from 2011 to 2016, and from 2016 to 2019 at p-value <0.05.

## Conclusion

The trend of minimum dietary diversity consumption among children aged 6 to 23 months in Ethiopia increased from 2011–2016 and decreased from 2016–2019 in the last decade. The study revealed that the changes in behavioral response and population composition contributed to MDD consumption change among children in Ethiopia. The finding highlights the urgent need for targeted interventions and policies to address the issue of MDD consumption change among young children due to population structure changes like wealth status, residence, and behavioral response related to employment, household head, and sex of the child in Ethiopia. Therefore, efforts should be geared to reduce poverty and improve maternal employment status, particularly for women, by producing equitable economic opportunities.

## Introduction

Adequate minimum dietary diversity (MDD) is defined based on the World Health Organization (WHO) and the United Nations Children's Fund (UNICEF). The child consumed at least five food groups from eight food groups like breast milk, grains roots and tubers, legumes and nuts, dairy products, flesh foods (meat, fish, poultry, and liver/organ meats), eggs, vitamin-A rich fruits and vegetables, and other fruits and vegetables [1]. Adequate MDD is crucial for the healthy growth and development of children aged 6 to 23 months [2].

After six months of age, children should receive adequate and acceptable supplemental foods while continuing to breastfeed. Nutrient requirements increase during this period, supporting overall health and development [2–4]. Childhood malnutrition due to inadequate feeding habits remains a significant challenge worldwide. It hinders long-term socioeconomic progress and poverty eradication efforts [5, 6]. In low and middle-income countries, insufficient dietary diversity contributes to rising mortality rates and the prevalence of diseases among children [7–9]. Notably, Sub-Saharan Africa (SSA) faces a high prevalence of inadequate MDD [10]. Insufficient nutrition adversely impacts children's prospects [11–13] and SSA bears the greatest burden of under-five mortality attributed to inadequate nutrition [14].

The Sustainable Development Goals (SDGs) aim to address these challenges by 2030 [15]. Specifically, SDG 2 focuses on ending hunger, achieving food security, and improving nutrition through universal access to safe and nutritious food to achieve SDG target 2.2 to end all forms of malnutrition [16]. However, in SSA mainly in Ethiopia, the situation remains contrary to these goals, with statistics showing a rise in malnutrition since 2015 [17].

Factors, such as the women's education [5, 18, 19], wealth index [5, 18, 19], antenatal care [20, 21], the husband's educational status [5, 18, 19, 21], child age [5, 22], mass media exposure

[5, 18, 19], maternal age [22, 23], occupational status [24, 25], and residence [25] were found to be associated factors of MDD among children aged 6 to 23 months in various literatures.

The existing epidemiological data does not conclusively establish whether the decline in MDD consumption resulted from the lack of intervention or changes in community demographics and behavior response. To bridge this gap in the literature, it is essential to assess the compositional and behavioral factors influencing MDD consumption.

### Theoretical framework on MDD consumption

In the context of child feeding practices and MDD consumption, the framework suggests that the trend of MDD consumption change is determined by the change in different factors. This includes composition change at the individual level like child characteristics (age, sex), maternal characteristics (age, education, occupation), composition change at the household characteristics (wealth status, household head sex, number of under-five living children, and household size), composition change at the community-level factors: residence, region, and behavioral change (coefficient) [19, 24–27] (Fig 1).

Then the trends and determinants of MDD consumption change found in this study can be interpreted through the lens of this theoretical framework that changes in individual, household, and community-level factors over time likely contributed to the observed trends in MDD consumption change. This analysis will yield reliable data for national policymakers. Furthermore, despite our exhaustive literature search, limited evidence exists regarding the trends and determinants of MDD consumption change among children aged 6 to 23 months over time in Ethiopia, specifically using decomposition analysis of data from the 2011–2019 EDHS. Although the decomposition analysis model is valuable for identifying the composition

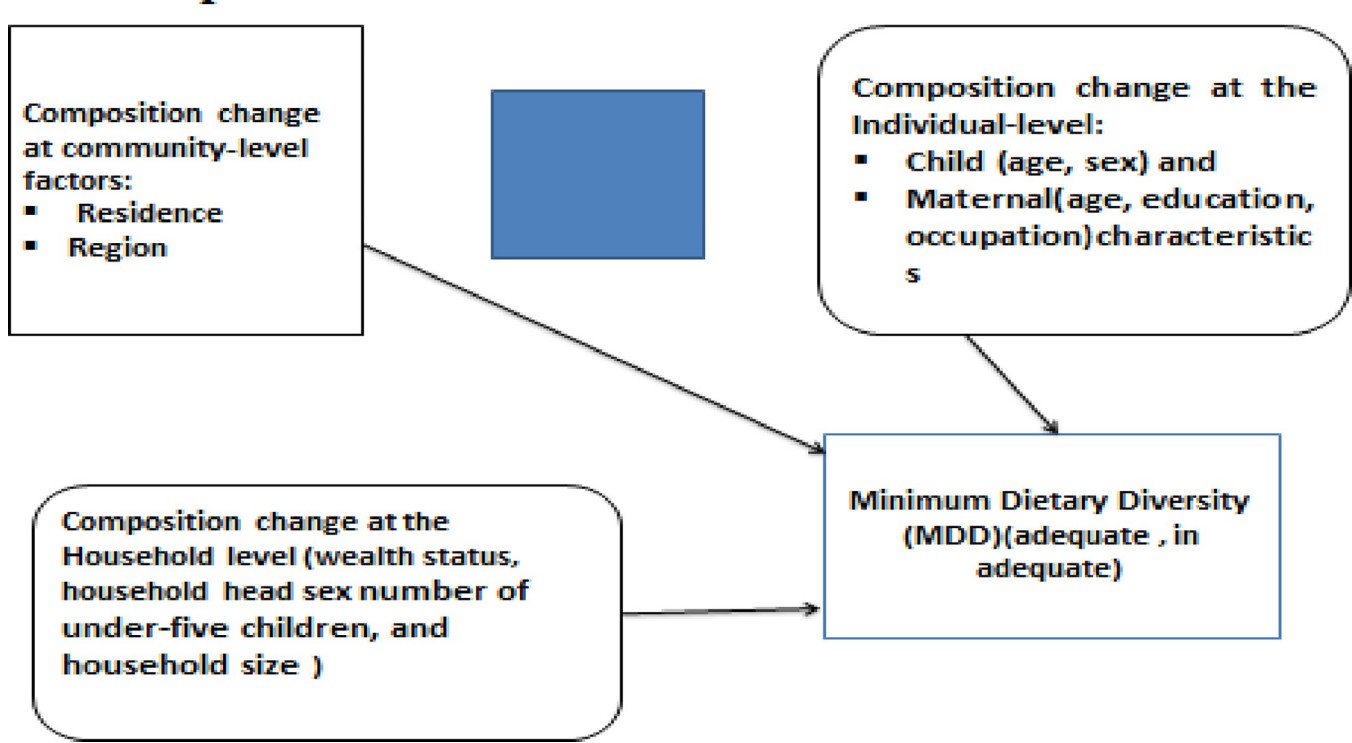

**Fig 1. Conceptual frameworks of determinants of MDD consumption in Ethiopia [19, 24–27].**

and behavior change factors contributing to changes in MDD consumption over time, it has not been explored in the context of Ethiopia. Therefore, the objective of this study was to analyze the trends and determinants of MDD consumption change among children aged 6 to 23 months in Ethiopia using data from the EDHS between 2011 and 2019.

## Methods

### Study design

A population-based cross-sectional study was used to investigate the trends in MDD consumption from 2011 to 2019 EDHS.

### Study setting

The research was conducted in Ethiopia, a country situated in East Africa with a diverse population. It shares borders with Sudan to the west, Somalia and Djibouti to the east, Eritrea to the north, and Kenya to the south. Ethiopia covers a total area of 1,112,000 square kilometers and is divided into eleven regions and two municipal governments. The Ethiopian landscape features a vast high plateau that spans the Rift Valley, extending into the northwest and southeastern highlands, each accompanied by its associated lowlands. The topographical variation is remarkable, ranging from 130 meters below sea level (in the Dallol depression located in the Afar region) to the peak of Mount Ras Dashen at 4,620 meters above sea level in the Semien Mountains [28].

**Population.** The source populations were all children aged 6–23 months and those in the selected Enumeration Areas (EAs) in each respective survey in Ethiopia were the study population.

### Data source

The data of this study were the DHS data from 2011–2019 EDHS and the data were extracted at https://dhsprogram.com/ by contacting them through personal accounts after justifying the reason for requesting the data [29]. The data set was birth record (BR) which contains the full birth history of all women interviewed. The data set also included information on pregnancy and postnatal care, as well as immunization, health, and nutrition like dietary food groups of children born in the last 5 years [30, 31]. The DHS was a household survey that was conducted every five years in low and middle-income countries. The data collected from the DHS survey was organized in a hierarchical structure, with households within a cluster forming the top level. The next level consists of household members, followed by interviewed women and men as a subset of household members. The bottom levels of the hierarchy include pregnancies and children of each interviewed woman [30, 31].

**Sample size and sampling method.** The study included representative weighted samples of 6,971, 7,467, and 3,154 children aged 6 to 23 months from the 2011, 2016, and 2019 rounds of the Ethiopian Demographic and Health Surveys (EDHS) respectively. Participants were selected using a two-stage stratified sampling technique. In the first stage, enumeration areas (EAs) were randomly chosen, followed by a random selection of households in the second stage.

### Variable measurement

A child who consumed at least five of the eight food groups in the 24 hours preceding the interview labeled as consuming adequate MDD"1" and else inadequate MDD "0" was the dependent variable [32]. Based on different literature and its public health significance of health policy development on nutrition: factors like; age of mother, father or child, sex of household head, educational status of mother and partner [5, 18, 19, 21], working status of

mother and father [24, 25], wealth index [5, 18, 19], residence [25], preceding birth interval, number of under-five living children, and household size were used as independent variable. In this study survey year was used as a decomposing variable which was coded as follows from 2011 to 2016 (2011 labeled as "0" and 2016 labeled as "1"), and from 2016 to 2019 (2016 labeled as "0" and 2019 labeled as "1").

## Data management, processing, and analysis

Data were combined using an append dataset and merged two datasets. Merging was conducted using cluster numbers. During the append dataset and merge of two datasets, the duplicate control was performed by creating a year code and duplicated values were dropped using the drop command in STATA version 17.0.

The consistency of data was checked using exploratory data analysis like frequency and percentage for categorical variables and summary statistics for continuous variables was conducted to identify outliers and missing values. Then we cleaned data by creating categories based on previous research criteria, dropping observations, dropping incomplete variables, dealing with outliers, and creating new variables for each survey. Variable generation, recoding, and labeling were conducted for the variable, which was needed. To account for the unequal probability of selection between the geographically defined strata and non-responses, we used sample weights. Survey design of complex sampling procedure was taken into account and it was declared using the SVY STATA command to control the clustering effect of complex sampling. The frequency with percentages for categorical variables was calculated to describe the characteristics of the study population by year of survey. The chi-square test ($X^2$) was used to examine the statistically significant change of MDD consumption from 2011 to 2016, and 2016 to 2019 EDHS. Before multivariate decomposition analysis first, bi-variable logistic regression was fitted and variables with p-values less than 0.25 were selected to develop multivariate decomposition analysis. Before we conducted multivariate decomposition Multicollinearity was checked among explanatory variables by using average standard error at a cut of point±2 and there is no Multicollinearity since the standard errors were between±2. To explain the contributing factors to the change in MDD consumption among children aged 6–23 months from 2011 to 2016, and 2016 to 2019 EDHS, a Multivariate decomposition (mvdcmp) analysis was conducted. Multivariate decomposition (mvdcmp) determines the high-outcome group automatically comparison group and uses the low-outcome group as a reference [33]. A multivariate decomposition analysis was conducted to investigate the source of change in MDD consumption from 2011 to 2016 and 2016 to 2019 EDHS among children aged 6–23 months in Ethiopia. The analysis used a logistic regression model to decompose the differences into two components. One that is explained by the difference in the population structure or composition across the survey. The second change was due to a change in the behavior of the survey population as the change in outcome was due to either a change in population composition or a change in behavior of the population. The observed differences in MDD consumption between different surveys were decomposed into characteristics (natural endowment or population composition) and a coefficient (effect of characteristics or behavioral effect). The logit-based difference can be decomposed as follows [33].

$$Y = F\left(\frac{e^{x\beta}}{1 + e^{x\beta}}\right) + \varepsilon \qquad (1)$$

$$Ya - Yb = F\left(\frac{e^{xa\beta a}}{1 + e^{xa\beta a}}\right) - F\left(\frac{e^{xb\beta b}}{1 + e^{xb\beta b}}\right) + \varepsilon \qquad (2)$$

$$\Delta Y = \left[ F\left( \frac{e^{xa\beta a}}{1 + e^{xa\beta a}} \right) - F\left( \frac{e^{xb\beta a}}{1 + e^{xb\beta a}} \right) \right] + \left[ F\left( \frac{e^{xb\beta a}}{1 + e^{xb\beta a}} \right) - F\left( \frac{e^{xb\beta b}}{1 + e^{xb\beta b}} \right) \right] + \varepsilon \qquad (3)$$

Where: Y is the dependent variable MDD consumption, X is the independent variable, β is the coefficient and F is the differential logistic function $X\left( \frac{e^{x\beta}}{1 + e^{x\beta}} \right)$ and Y: Hence, the result focused on how MDD consumption responded to population composition and their behavior and how these factors shape it across different surveys at different times and ε is the row difference. The level of statistical significance was set at a p-value of less than 0.05.

**Ethical approval.** No ethical approval was needed because we had used the demographic and health survey, which identifies all data before making it public, and the used DHS data sets are openly accessible. An authorization letter was requested to download the DHS data set and this was obtained from the Central Statistical Agency (CSA) after being requested at https://dhsprogram.com/. The dataset and all methods of this study were conducted according to the guidelines laid down in the Declaration of Helsinki and based on DHS research guidelines.

## Result

### The socio-demographic characteristics of respondents from 2011–2019

Table 1 presents the trends of socio-demographic characteristics of respondents for MDD among children aged 6–23 months in Ethiopia from 2011 to 2016, and 2016 to 2019 based on weighted data. The table provides information on various variables and their frequencies and percentages for each year of the survey. The percentage of mothers aged 25–34 slightly increased from 39% to 43% and mothers aged 35–49 also slightly increased from 41% to 42% from 2011 to 2016. However, the percentage of mothers aged 25–34 decreased from 43% to 18% and mothers aged 35–49 also decreased from 42% to 12% from 2016 to 2019. The percentage of households with more than five members slightly increased from 40% to 42% in 2011 to 2016. However, the percentage of households with less than five members was slightly decreased from 45% to 21% in 2016 to in 2019. The percentage of male household heads increased from 40% to 42% from 2011 to 2016 and decreased from 42% to 18% from 2016 to 2019. The percentage of households classified as middle and rich slightly increased from 40% to 44% and 41% to 42% in 2011 to 2016 and decreased from 44% to 16% and 42% to 17% in 2016 to 2019 respectively. The percentage of children living in households in rural areas increased from 40% to 44% from 2011 to 2016. However, the proportion of children living in households in urban areas increased from 27% to 36% from 2016 to 2019. The percentage of male children decreased from 49% to 18% in 2016 to 2019. The percentage of birth intervals less than or equal to 24 months increased from 37% to 45% from 2011 to 2016 but decreased from 45% to 18% from 2016 to 2019. The percentage of children aged 6–11 months increased from 40% to 43% in 2011 to in 2016. The percentage of children aged 12–24 months decreased from 42% to 19% in 2016 to 2019.

### Trends of MDD consumption in Ethiopia

In the study period (2011–2019), the consumption of adequate MDD increased from 2% in 2011 to 10.41% in 2016 but decreased from 10.41% in 2016 to 7.11% in 2019 in Ethiopia.

Fig 2 illustrates the trends of MDD consumption in Ethiopia. In Ethiopia, the consumption of adequate MDD among children was highest in 2016 (10.41%), and lowest in 2011 (2%). The incremental change of adequate MDD consumption from 2011–2016 was 8.41% and the decrement change from 2016–2019 (3.3%). The changes in MDD consumption among children

**Table 1. Trends of socio-demographic characteristics of the respondents for MDD among children aged from 6–23 months in Ethiopia from 2011–2019 EDHS (weighted).**

| Variables (Characters) | Year of Survey | | | |
|---|---|---|---|---|
| | **2011(n = 6971)** | **2016(n = 7467)** | **2019(n = 3154)** | **Total(n = 17592)** |
| | **Frequency(percent)** | **Frequency(percent)** | **Frequency(percent)** | **Frequency(percent)** |
| Maternal age | | | | |
| 15–24 | 155(35%) | 191(43%) | 97(22%) | 443 |
| 25–34 | 3428(39%) | 3810(43%) | 1636(18%) | 8874 |
| 35–49 | 3388(41%) | 3466(42%) | 1421(17%) | 8275 |
| Marital status | | | | |
| Un-union | 285(52%) | 194(36%) | 65(12%) | 544 |
| Union | 6685(39%) | 7274(43%) | 3089(18%) | 17048 |
| Educational status of a mother | | | | |
| Have no education | 5340(39%) | 6059(44%) | 2276(17%) | 13675 |
| Have education | 1631(42%) | 1408(36%) | 878(22%) | 3917 |
| Educational status of the father | | | | |
| Have no work | 4673(46%) | 5544(54%) | | 10217 |
| Have work | 2282(54%) | 1924(46%) | | 4206 |
| Sex of household head | | | | |
| Male | 6199(40%) | 6627(42%) | 2786(18%) | 15611 |
| Female | 772(39%) | 840(42%) | 369(19%) | 1981 |
| Household size | | | | |
| <5 | 912(37%) | 1063(43%) | 517(21%) | 2492 |
| >5 | 6059(40%) | 6404(42%) | 2638(17%) | 15101 |
| Number of U5 children | | | | |
| <4 | 6870(40%) | 7274(42%) | 3115(18%) | 17259 |
| >4 | 101(30%) | 194(58%) | 39(18) | 333 |
| Household wealth index | | | | |
| Poor | 3294(39%) | 3573(42%) | 1667(20%) | 8534 |
| Middle | 1495(40%) | 1654(44%) | 606(16%) | 3755 |
| Rich | 2182(41%) | 2240(42%) | 882(17%) | 5303 |
| Residence | | | | |
| Urban | 657(38%) | 459(27%) | 615(36%) | 1731 |
| Rural | 6314(40%) | 7008(44%) | 2539(16%) | 15861 |
| Sex of child | | | | |
| Male | 3403(39%) | 3655(42.4%) | 1563(18.12%) | 8622 |
| Female | 3568(40%) | 3812(42%) | 1591(18%) | 8971 |
| Twin(types) of Birth | | | | |
| Single | 6868(40%) | 7411(43%) | 3104(18%) | 17383 |
| Multiple | 103(49%) | 56(27%) | 519(24%) | 209 |
| Preceding Birth interval | | | | |
| < = 24month | 1,722(37%) | 2,117(45%) | 858(18%) | 4,697 |
| >24Month | 3,276(42%) | 3,222(41%) | 1,340(17%) | 7,837 |
| Child age | | | | |
| 6–11 Month | 4,327(40%) | 4,600(43%) | 1,888(17%) | 10,815 |
| 12–23 month | 2,644(39%) | 2,867(42%) | 1,267(19%) | 6,777 |

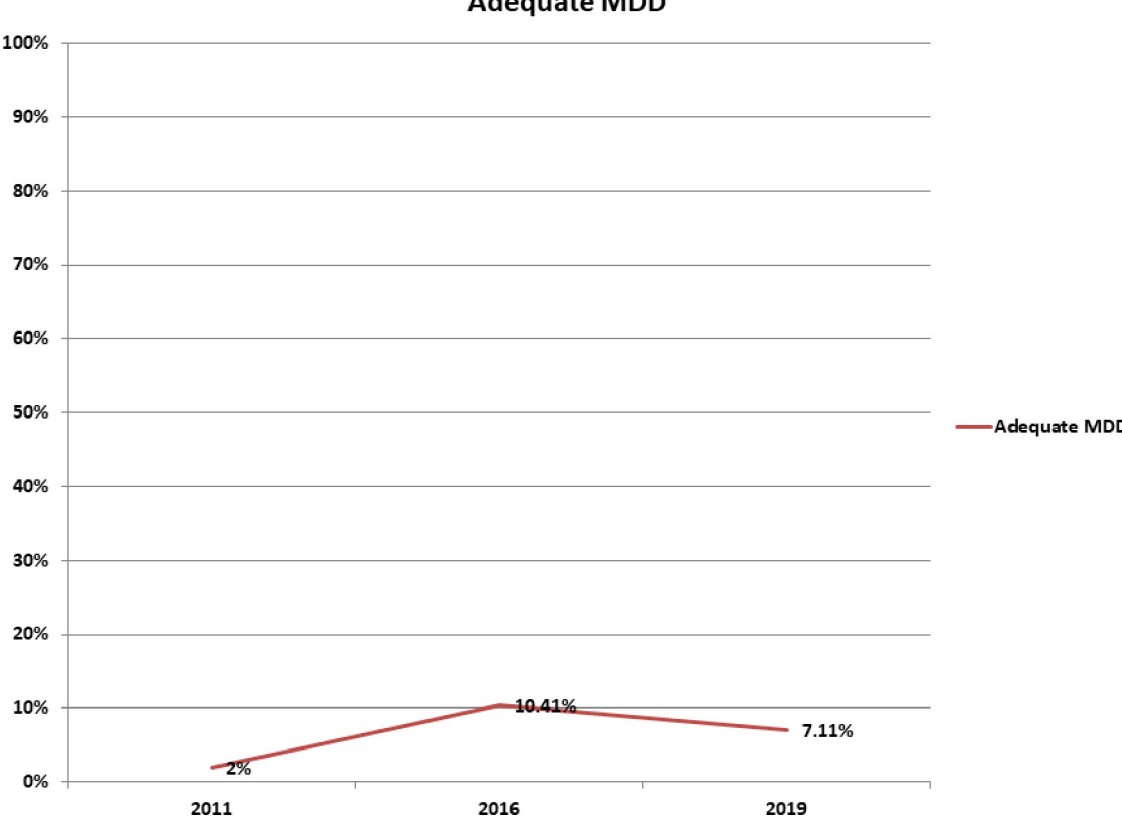

**Fig 2. Trends of MDD consumption among children aged 6–23 months from 2011 to 2019 using Ethiopian Demographic and Health Survey Data.**

were highly statistically significant from 2011–2016, and 2016–2019 (Chi-square test, p-value <0.001) (Table 2).

## Multivariate decomposition of analysis of MDD consumption change from 2011–2016

A decomposition analysis model was fitted to account for differences in characteristics (compositional factors) and differences due to the effect of factors (behavioral characteristics) for change in MDD consumption.

**Table 2. Chi-square test for change in minimum dietary diversity consumption among children from 2011 to 2016 and 2016 to 2019 in Ethiopia.**

|  | In adequate MDD | adequate MDD |
|---|---|---|
| 2011 | 6325 | 119 |
| 2016 | 5990 | 508 |
| Chi-square result | Pearson chi2($x^2$) = = 250.2332 P-value = 0.000 | |
| 2016 | 5990 | 508 |
| 2019 | 2947 | 164 |
| Chi-square result | Pearson chi2($x^2$) = 20.9688 P-value = 0.000 | |

Key: chi2($x^2$) = chisquare

The overall incremental change in the consumption of adequate MDD among children aged 6–23 months in Ethiopia between 2011 and 2016 was 8.41%.

## Changes due to characters (Endowment) and the effect of characters (Coefficient)

According to a multivariate decomposition analysis, the differences in the compositional characteristics (population demographic change) of children accounted for 4.8% of the overall change in consumption of adequate MDD. However, none of the compositional character changes has a significant relationship with changes in the consumption of adequate MDD.

After controlling the effect of compositional factors (population structure change), around 104.8% of the overall change in consumption of adequate MDD was due to behavioral characteristics (unexplained factors like positive or negative behavior change). In multivariate decomposition analysis of 2011–2016 EDHS data sets, after controlling the effect of compositional factors, behavioral change factors like; maternal age, wealth index, sex of household head, and paternal occupational status, had a statistically significant contribution to change in the consumption of adequate MDD (Table 3). The behavior related to the higher age of mothers aged 35–49 contributed to an increasing 45% change in consumption of adequate MDD from 2011 to 2016. The negative coefficient showed a negative relationship with the change in consumption of adequate MDD among children aged 6–23 months in Ethiopia. The maternal occupation contributed by increasing 23.9% change in consumption of adequate MDD increment from 2011 to 2016. However, fathers' working status contributed by decreasing the 42.2% change in consumption of adequate MDD from 2011 to 2016. The female household head had a negative contribution by reducing the 5.3% change in consumption of adequate MDD from 2011 to 2016. Children from the middle wealth index contributed by increasing 7.02%% change in consumption of adequate MDD from 2011 to 2016 (Table 3).

**Multivariate decomposition of analysis of MDD consumption change from 2016–2019.** We found that 3.3% decrement change in the consumption of adequate MDD among children aged 6–23 months in Ethiopia from 2016 and 2019.

**Changes due to characters (Endowment) and the effect of characters (Coefficient).** The 16.89% of changes in the consumption of adequate MDD were explained due to changes in composition characters. Among the compositional factors: wealth index, residence, and sex of the child had made a significant contribution to the change in consumption of adequate MDD (Table 4). Composition change in the household wealth index (the percentage of households classified as middle and rich decreased from 44% to 16% and 42% to 17% from 2016 to 2019) contributed to the reduction of adequate MDD consumption by 7.39% and 2.95% respectively. Due to the composition change in population and place of residence, the decrease of rural dwellers from 44% to 16% and the increment of urban dwellers from 27% to 36% from 2016 to 2019 respectively, the less density rural residence contributed by reducing (preventing) 25.17% of MDD consumption reduction from 2016 to 2019.

After controlling the effect of compositional factors, we found that 116.89% change in MDD consumption between 2016 and 2019 among children aged 6–23 months in Ethiopia accounted for behavioral change factors. Behavioral factors such as household wealth index, rural residence, and sex of the child made a significant contribution to the change in consumption of adequate MDD from 2016 to 2019. Children from households with rich wealth index contributed by decreasing 59.9% changes in adequate MDD consumption reduction from 2016 to 2019. A child who was female sex had a 35.57% negative impact on the reduction of changes in adequate MDD consumption between 2016 and 2019.

**Table 3. Multivariate decomposition analysis of minimum dietary diversity consumption change among children aged 6–23 months from 2011–2016 in Ethiopia using demographic and health survey data.**

| Minimum dietary diversity | Coef. | P-value | 95% CI | Interval] | Percent | | | | | |
|---|---|---|---|---|---|---|---|---|---|---|
| E | -0.004 | 0.000 | -0.006 | 0.002 | -4.830 | | | | | |
| C | 0.089 | 0.000 | 0.081 | 0.097 | 104.830 | | | | | |
| R | 0.085 | 0.000 | 0.078 | 0.093 | | | | | | |
| **Decomposition** | **Differences Due to Characters (Endowment)** | | | | | **Differences Due to the Effect of Characters (Coefficient)** | | | | |
| Minimum dietary diversity | Coef. | P-value | 95% Conf. of Coef. | | Percent | Coef. | P-value | 95% Conf. of Coef. | | Percent |
| Maternal age | | | | | | | | | | |
| 15–24 | Refer. | | | | | | | | | |
| 25–34 | -0.0002 | 0.76700 | -0.0017 | 0.00126 | -0.26000 | 0.01630 | 0.14400 | -0.0056 | 0.03815 | 19.15000 |
| 35–49 | -0.0002 | 0.79500 | -0.0019 | 0.00147 | -0.27000 | 0.03840 | 0.00100 | 0.01565 | 0.06114 | 45.11000 |
| Educational status of the mother | | | | | | | | | | |
| Have no education | Refer. | | | | | | | | | |
| Have education | -0.0012 | 0.43500 | -0.0043 | 0.00183 | -1.43000 | -0.0042 | 0.11600 | -0.0093 | 0.00102 | -4.89000 |
| Occupation status mother | | | | | | | | | | |
| Have no work | Refer. | | | | | | | | | |
| Have work | 0.01137 | 0.42900 | -0.0168 | 0.03954 | 13.35000 | 0.02035 | 0.00001 | 0.01159 | 0.02911 | 23.91000 |
| Marital status | | | | | | | | | | |
| Un-union | Refer. | | | | | | | | | |
| Union | -0.0095 | 0.24000 | -0.0253 | 0.00632 | -11.120 | 0.11345 | 0.29700 | -0.0998 | 0.32678 | 133.29000 |
| Occupation status father | | | | | | | | | | |
| Have no work | Refer. | | | | | | | | | |
| Have work | -0.0061 | 0.42500 | -0.0210 | 0.00888 | -7.17000 | -0.0361 | 0.1 | -0.0495 | 0.0227 | -42.41000 |
| Sex of household head | | | | | | | | | | |
| Male | Refer. | | | | | | | | | |
| Female | 0.00011 | 0.45700 | -0.0002 | 0.00039 | 0.13000 | -0.0048 | 0.01100 | -0.0077 | -0.0009 | -5.13000 |
| Household wealth index | | | | | | | | | | |
| Poor | Refer. | | | | | | | | | |
| Middle | -0.0009 | 0.42400 | -0.0030 | 0.00128 | -1.03000 | 0.00597 | 0.02500 | 0.00075 | 0.01119 | 7.02000 |
| Rich | 0.00089 | 0.42500 | -0.0013 | 0.00306 | 1.04000 | 0.0050 | 0.20600 | -0.0028 | 0.01296 | 5.97000 |
| Residence | | | | | | | | | | |
| Urban | Refer. | | | | | | | | | |
| Rural | 0.00239 | 0.45200 | -0.0039 | 0.00863 | 2.81000 | 0.01250 | 0.36700 | -0.0147 | 0.03970 | 14.69000 |
| Sex of the child | | | | | | | | | | |
| Male | Refer. | | | | | | | | | |
| Female | -0.00004 | 0.44500 | -0.0001 | 0.00006 | -0.04000 | -0.0010 | 0.82700 | -0.0103 | 0.00826 | -1.22000 |
| Twin birth | | | | | | | | | | |
| Single | Refer. | | | | | | | | | |
| Multiple | -0.00075 | 0.47000 | -0.0028 | 0.00129 | -0.89000 | 0.00091 | 0.52900 | -0.0019 | 0.00374 | 1.07000 |
| Age of child | | | | | | | | | | |
| 6–11 month | Refer. | | | | | | | | | |
| 12–23 month | 0.00003 | 0.65200 | -0.0001 | 0.00018 | 0.04000 | -0.0057 | 0.14600 | -0.0134 | 0.00199 | -6.72000 |

**Key:** Refer. = Reference Coef. = coefficient, Pct. = percent, 95%Conf. of Coef. = 95% confidence interval of the coefficient, E = endowment, C = coffiecent, R = row difference

**Table 4. Multivariate decomposition analysis of minimum dietary diversity consumption change among children aged 6–23 months from 2016–2019 in Ethiopia using demographic and health survey data.**

| Minimum Dietary Diversity | Coef. | P-value | [95% Conf. | | Percent | | | | | |
|---|---|---|---|---|---|---|---|---|---|---|
| E | -0.01 | 0.04 | -0.01 | 0.00 | -16.89 | | | | | |
| C | 0.04 | 0.00 | 0.03 | 0.05 | 116.89 | | | | | |
| R | 0.03 | 0.00 | 0.02 | 0.04 | | | | | | |
| **Decomposition** | **Differences Due to Characters (Endowment)** | | | | | **Differences Due to the Effect of Characters (Coefficient)** | | | | |
| Minimum Dietary Diversity | Coef. | P-value | 95% Conf. of Coef. | | Percent | Coef. | P-value | 95% Conf. of Coef. | | Percent |
| Maternal age | | | | | | | | | | |
| 15–24 | Refer. | | | | | | | | | |
| 25–34 | -0.00008 | 0.69800 | -0.0005 | 0.00032 | -0.2400 | 0.00984 | 0.54600 | -0.022 | 0.04179 | 29.92000 |
| 34–49 | -0.00015 | 0.65500 | -0.0008 | 0.00051 | -0.4600 | -0.0174 | 0.22600 | -0.0455 | 0.01076 | -52.89000 |
| Educational status of the mother | | | | | | | | | | |
| No education | Refer. | | | | | | | | | |
| Have education | 0.0016 | 0.0870 | -0.0002 | 0.0034 | 4.8000 | -0.0032 | 0.3530 | -0.0099 | 0.0036 | -9.7100 |
| Marital status | | | | | | | | | | |
| Un-union | Refer. | | | | | | | | | |
| Union | -0.0021 | 0.257 | -0.035 | 0.07 | -6.4200 | 0.4115 | 0.0890 | -0.1009 | 0.7221 | 1251.4500 |
| Sex of household head | | | | | | | | | | |
| Male | Refer. | | | | | | | | | |
| Female | 0.00014 | 0.05700 | 0.00001 | 0.00028 | 0.42000 | 0.00058 | 0.80200 | -0.0039 | 0.00510 | 1.75000 |
| Household wealth index | | | | | | | | | | |
| Poor | Refer. | | | | | | | | | |
| Middle | 0.00243 | 0.0001 | 0.00119 | 0.00367 | 7.39000 | -0.0022 | 0.39200 | -0.0072 | 0.00282 | -6.66000 |
| Rich | 0.00097 | 0.00100 | 0.00041 | 0.00153 | 2.95000 | -0.0197 | 0.00001 | -0.0263 | -0.0131 | -59.90000 |
| Residence | | | | | | | | | | |
| Urban | Refer. | | | | | | | | | |
| Rural | -0.00827 | 0.01300 | -0.0148 | -0.0017 | -25.170 | -0.0497 | 0.00001 | -0.0773 | -0.0221 | -151.1600 |
| Sex of child | | | | | | | | | | |
| Male | Refer. | | | | | | | | | |
| Female | -0.00011 | 0.02700 | -0.0002 | -0.0001 | -0.3500 | -0.0117 | 0.0270 | -0.0220 | -0.0012 | -35.57000 |
| Age of child | | | | | | | | | | |
| 6–11 month | Refer. | | | | | | | | | |
| 12–23 month | 0.00006 | 0.67000 | -0.0002 | 0.00034 | 0.18000 | 0.00728 | 0.09800 | -0.0012 | 0.01592 | 22.15000 |

**Key:** Refer. = Reference Coef. = coefficient, Pct. = percent, 95% Conf.of coef = 95% confidence interval of the coefficient E = endowement,C = coffiecent, R = row difference

## Discussion

Analyzing national-level trends, composition, and behavioral factors that impact minimum dietary diversity can offer valuable insights for creating contextually relevant strategies and policies. Additionally, this information serves as evidence in support of SDG 2 target 2.2, which aims to eradicate all forms of malnutrition. Therefore, emphasizing trends in recommended minimum dietary diversity and understanding contributing factors for enhancing children's nutrition is crucial for developing comprehensive interventions to improve child health and nutritional well-being. Therefore, this study was conducted to determine trends and determinants of minimum dietary diversity (MDD) consumption change among children aged 6 to 23 months in Ethiopia from 2011–2019 EDHS using trend and multivariate decomposition analysis. The trends of adequate MDD consumption among children aged 6 to 23

months in Ethiopia significantly increased from 2011–2016 but decreased from 2016–2019. This was supported by a previous study finding in SSA [25, 34].

The overall trend of adequate (MDD) consumption among children in the last decade was higher than study findings in the Afar region [35] and Dangle District in Ethiopia [36] however, lower than study studies conducted in Wolaita Sodo town [37], Debre Tabor Town [38], Bale Zone [39], and Dabat District [40]. This discrepancy may arise due to sample size differences and may arise because of environmental changes, socioeconomic variation, and differences in healthcare utilization [41]. It is also important to note that the discrepancy in the proportion of MDD consumption among children might be due to unequal attention given to infant and young child feeding by regions, which is precipitated by pre-existing underlying factors such as drought and political instability [41–43]. The reduction of adequate (MDD) consumption among children from 2016 to 2019 in the current study was due to the composition change of households classified as middle and rich decreased from 2016 to 2019. This may also be due to rapid population growth, socioeconomic crises, drought, and other artificial and natural occurrences that have direct and indirect effects on population change and food supply in Ethiopia [41–43].

Changes in population composition were one of the determinant variables in a positive and negative direction of change in (MDD) consumption among children in Ethiopia. This implied that a significant contribution to the change in (MDD) consumption among children was due to the change in population composition as evidenced by the descriptive result. According to multivariate decomposition, none of the compositional factors contributed to the change in (MDD) consumption among children from 2011–2016, and compositional change factors (such as wealth index, residence, sex, and age of child significantly contributed to the change in (MDD) consumption among children from 2016–2019 in Ethiopia. Changes in the household wealth index, with a decrease in the percentage of households classified as middle and rich from 2016 to 2019, contributed to a reduction in adequate MDD consumption from 2016 to 2019. This is in line with study findings in Addis Ababa, Ethiopia [44], Southern Ethiopia [20], Indonesia [22], Southern Africa [18], Bangladesh [23]. This may be due to children from poor households may not have access to eggs, fruits, and vegetables due to economic constraints. As a result, they may be less likely to consume a diverse range of foods. To alleviate their financial difficulties, they may buy less expensive food items. Unfortunately, many children are unable to consume fruits and vegetables due to economic constraints evidenced by a study conducted in Ethiopia [45]. From 2016 to 2019, the number of children in rural areas decreased while in urban increased. As a result, rural residence helped to reduce the changes in MDD consumption reduction from 2016 to 2019, which was supported by descriptive analysis. Even though it contradicted the previous study findings in Ethiopia [44], Sri Lanka [46]. This may due to increasing urban population density affecting the ability to obtain food with limited income in the face of food shortages caused by poor linkage between urban and rural markets and pricing by sellers. This may also be because rural communities are more agrarian, which allows them to easily access a variety of foods without limitations due to their low population density.

After controlling the effect of compositional factors, behavioral change factors like maternal age, occupational status of parents, sex of the household head, wealth index, residence, and sex of child significantly contributed to the change in (MDD) consumption among children across surveys. The consumption of adequate MDD increased from 2011 to 2016 and was attributed to the behavior of mothers aged 35–49. This may be because as the mother's age increases; they may gain experiential knowledge (positive behavior) that helps them feed their children adequately. The mothers behavior of who were employed contributed to an increased change in the consumption of adequate MDD from 2011 to 2016. This is supported by other study

findings in East Gojjam Ethiopia [27], Central Ethiopia [21], and Bangladesh [23]. The behaviors of the female household head had a negative contribution to the change in consumption of adequate MDD from 2011 to 2016. This could be due to women having less decision-making power and mobility, which affects their ability to go to the marketplace and purchase food. This is a common issue in many households where women are not given equal opportunities to make decisions and have limited mobility. This can lead to a lack of access to healthy food options, which can have negative impacts on the health of children. It is important to address this issue by empowering women and providing them with the resources, they need to make informed decisions about their family's health and well-being. The behaviors of mothers from middle-wealth index households contributed by increasing change in consumption of adequate MDD among children from 2011 to 2016 and the behavior of mothers from rich-wealth index households contributed by decreasing changes in adequate MDD consumption reduction from 2016 to 2019. This is supported by other study findings in East Gojjam Ethiopia [27], Central Ethiopia [21], Sri Lanka [46], and Bangladesh [23]. This may be related to children from middle and rich households being more likely to get eggs, fruits, and vegetables because of the economic well-being of the family.

Strength and limitation of the study: This study's use of nationally representative data, which allows it to be generalizable to all children in Ethiopia could be a strength of the study. The DHS is conducted every five years and may not capture real-time changes or trends in child health and nutritional issues since these surveys were not collected among the same participants could be the limitation of the study. The data collected through DHS rely on self-reported information, which can be subject to recall bias or social desirability bias could also be the limitation of this study.

## Conclusion

The trend of minimum dietary diversity (MDD) consumption among children aged 6 to 23 months in Ethiopia increased from 2011–2016 but decreased from 2016–2019 in the last decade. The changes in behavioral response related to increasing maternal age, employed mothers, and better wealth status positively and female household heads negatively contributed to the improvement of MDD consumption from 2011–2016. The population composition change in wealth index positively and rural place of residence and female sex of the child negatively contributed to the reduction of MDD consumption from 2016 to 2019. The behavioral response change being from a rich household, living in a rural residence, and being a female child negatively contributed to the reduction of MDD consumption from 2016 to 2019. Therefore, efforts should be geared to reduce poverty and improve maternal employment status, particularly for women, promoting equitable economic opportunities. Special interventions may be required to promote the consumption of MDD, such as vouchers to help with the purchase of these foods for poor communities. These interventions can be provided by various nutrition organizations, such as the safety net program.

## Acknowledgments

The authors are sincerely grateful to the Demographic Health Survey (DHS) program for providing us to use the EDHS dataset through their archives (https://dhsprogram.com)

## Author Contributions

**Conceptualization:** Abel Endawkie, Yawkal Tsega.

**Data curation:** Abel Endawkie.

**Formal analysis:** Abel Endawkie.

**Investigation:** Abel Endawkie.

**Methodology:** Abel Endawkie.

**Software:** Abel Endawkie.

**Supervision:** Abel Endawkie.

**Validation:** Abel Endawkie.

**Visualization:** Abel Endawkie.

**Writing – original draft:** Abel Endawkie, Lakew Asmare.

**Writing – review & editing:** Abel Endawkie, Hiwot Tadesse Alemu, Demiss Mulatu Geberu, Asebe Hagos, Melak Jejaw, Kaleab Mesfin Abera, Misganawu Guadie Tiruneh, Kaleb Assegid Demissie, Yawkal Tsega, Adina Yeshambel Belay, Wubshet Debebe Negash, Amare Mesfin Workie, Lamrot Yohannes, Mihret Getnet, Nigusu Worku.

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
