## [Decision Letter · Decision Letter 0]

5 Mar 2024

PONE-D-24-02192Change in Minimum Dietary Diversity Consumption and Its Determinants among Children Aged 6-23 months in Ethiopia using EDHS 2011 to 2019: A Multivariate Decomposition AnalysisPLOS ONE

Dear Dr. Endawkie,

Thank you for submitting your manuscript to PLOS ONE. After careful consideration, we feel that it has merit but does not fully meet PLOS ONE’s publication criteria as it currently stands. Therefore, we invite you to submit a revised version of the manuscript that addresses the points raised during the review process.

We look forward to receiving your revised manuscript.

Kind regards,

Satyajit Kundu

Academic Editor

PLOS ONE

Additional Editor Comments:

The authors are requested to address the reviewers' comments carefully and in detailed.

Reviewers' comments:

Reviewer's Responses to Questions

**Comments to the Author**

1. Is the manuscript technically sound, and do the data support the conclusions?

Reviewer #1: Partly

Reviewer #2: Yes

2. Has the statistical analysis been performed appropriately and rigorously? 

Reviewer #1: No

Reviewer #2: Yes

3. Have the authors made all data underlying the findings in their manuscript fully available?

Reviewer #1: Yes

Reviewer #2: Yes

4. Is the manuscript presented in an intelligible fashion and written in standard English?

Reviewer #1: No

Reviewer #2: Yes

5. Review Comments to the Author

Reviewer #1: I would like to thank the editor for giving me opportunity to review this article. Here are few points I am concerned about.

Methodology

1. Section: Independent variables- Why have you decided to choose these variables? Adding necessary details along with references may help.

2. Section: Data processing and analysis – Will it be possible to explain how authors did data cleaning, checked data consistency, and variables recoding in detail?

3. Section: Data processing and analysis – The authors are using multivariate decomposition analysis. Will it be possible to describe the underlying methodology behind this in detail? I believe this description will help in understanding what this methodology is doing underneath.

4. Overall, will it be possible to merge some of the sub-sections? Such as sample size and data management section may be discussed in data processing and analysis.

5. Some sentences are not worded right?

Results

1. Section: Change in MDD consumption in Ethiopia- Line 196 – 197, I do not find table showing chi-square test. Will it be possible to include it? Same applies for line 203.

2. Line 218 – 219: Authors are claiming around 104.8% was due to the behavioral characteristics. Is this meaningful?

3. How did authors calculate percentage of contribution? Description needed.

4. In table 2, I see some of the coefficients are zero. I am assuming the coefficients are close to zero. Since authors reported up to one digit after decimal, I am seeing it zero. The associated 95% confidence interval is also (0.0, 0.0). Such as for category rural of variable residence. But I am seeing the percentage of contribution is 14.7%. Will it be meaningful to refit this model with deleting variables with coefficients close to zero at least which are not significant?

5. Revising presentation on table 2.

6. Comment 2 applies on line 247.

7. Comment 4 applies on table 3.

8. From table 3, I see, the percentage of increase due to the category union of marital status is 1251.45% and the percentage of reduction due to the category rural of variable residence is 151.160%. Second one is significant. Are they meaningful?

9. Will it be meaning to describe how difference due to characters and due to the effect of characters differ?

Reviewer #2: Thank you for the opportunity to review this study. The authors attempted to o determine trend of MDD and its determinants among children aged 6 to 23 months in Ethiopia from 2011-2019. While the study is good, I have some comments to strengthen the paper.

Abstract

1. Line 45: Specify the particular SDG target.

2. Line 53: Did you do a trend analysis or a change analysis?

3. Line 61: Should be 'p-value' not 'P-value'

4. Line 63: Change 'wasn't' to 'was not'.

5. The authors must rewrite the conclusion section. Lines 66-68 are just a repetition of the findings. At least, we should have a glimpse of the specific, practical recommendations in the conclusion.

Background

6. The paragraphs in the background are too long which makes it difficult to get the main idea. Kindly restructure your paragraphs. A paragraph should contain one central theme.

7. In the first paragraph, the authors present issues relating to dietary insufficiency. However, they fail to explain what this insufficiency is. There are no standards presented to show at what threshold a diet will be deemed as insufficient. This should be presented.

8. I suggest a complete overhaul of the background. Tell the story in a coherent manner. Begin from the broader perspective looking at the global, regional and national statistics on dietary insufficiency in children. Then you can zoom in to discuss its adverse health effect so as to allow reader to appreciate the importance of the study. This topic has been extensively researched and so we know about the determinants already. So, you must present what has been done in that regard and tell us where the gap is - the changes in MDD.

Methods

9. The methodology is scanty. The authors have to discuss the design, and sampling in detail. Make sure you cite appropriately as this information are from other sources.

10. Line 154: What do the authors mean by the sampling design was declared? This should be articulated clearly.

Results and discussion

11. Did you account for collinearity? Line 260: Correct the typographical error.

12. Write '6-23 months' not 'six to 23 months'.

13. Lines 260-262: The authors stated, "This suggests that children in Ethiopia between the ages of six and twenty-three months have not been getting minimum dietary diversity and are not in good progress." This argument is too superficial. You need to discuss in detail what might have accounted for the increment seen between 2011-2016, and the decreased observed for 2016-2019. What are the plausible explanations?

14. The authors argue that drought and political instability accounted for the decrease but there are no references for us to confirm this. How did that happen? Are there evidence to support this argument?

15. Why do the authors cite mainly studies from South East Asia when this study is conducted in an SSA country? There are a number of studies conducted in many SSA countries that are similar to the current study. You should note that there are sociocultural differences that affects such comparisons. The authors should be mindful of that and compare their study with other SSA countries.

16. Given the nature of this study, it will be advisable for the authors to include a theoretical framework to strengthen their findings and lay the foundation for their arguments in the discussion section.

17. Again, the paragraphs are too long which makes it difficult to grasps the discussion. Discuss one finding in one paragraph. Also, the strengths and limitations should be a new paragraph.

Conclusion

18. See the comment on the conclusion as stated in the abstract.

Line 366: It should be 'funding' not 'founding'.

6. PLOS authors have the option to publish the peer review history of their article (what does this mean?). If published, this will include your full peer review and any attached files.

Reviewer #1: No

Reviewer #2: No

---

## [Author Response · Author response to Decision Letter 0]

13 Mar 2024

Authors response to Reviewer Comment 

Reviewer#1 Comments

General comments by Reviewer #1: I would like to thank the editor for giving me the opportunity to review this article. Here are a few points I am concerned about.

Response: We are grateful to thank you for your kindness and response. We have tried to correct your concern and suggestion thoroughly.

Methodology

Comment 1: Independent variables- Why have you decided to choose these variables? Adding necessary details along with references may help.

Response 1: We sincerely appreciate your thoughtful feedback. After careful review, we made the necessary revisions and corrections. As part of our standard practice, we chose relevant variables by reading various literature sources related to the factors of minimum dietary diversity consumption among children aged 6 to 23 months in Ethiopia. These independent variables were selected based on insights from different research, alignment with sustainable development goals, and adherence to national policy and nutritional guidelines. We also referenced relevant sources that informed our decisions.

Comment 2: Data processing and analysis – Will it be possible to explain how authors did data cleaning, checked data consistency, and variables recoding in detail?

Response 2: We extend our sincere gratitude for your insightful comment. We found it particularly valuable during the writing process of our work. To ensure data consistency, we conducted exploratory analyses, including frequency and percentage distribution, as well as summary statistics for each variable across all surveys. Our consistency checks involved identifying outliers and missing values. Subsequently, we meticulously cleaned the data by categorizing variables based on established guidelines and previous research criteria. We addressed outliers and dropped incomplete observations. Additionally, we performed variable generation, recording, and labeling as needed from the existing variables of the data. For instance, the outcome variable was generated from eight distinct food groups: 1. breast milk, 2. grains, roots, and tubers, 3. legumes and nuts, 4. dairy products, 5. flesh foods (meat, fish, poultry, and liver/organ meats), 6. Eggs, 7. vitamin-A-rich fruits and vegetables, and 8. other fruits and vegetables. These were categorized into two groups: adequate and inadequate minimum dietary diversity (MDD).” Similarly to this, important variables were generated, recorded, and labeled as needed.

Comment3: Data processing and analysis – The authors are using multivariate decomposition analysis. Will it be possible to describe the underlying methodology behind this in detail? I believe this description will help in understanding what this methodology is doing underneath.

Response 3: We sincerely appreciate your thoughtful comment. In response to your concern, we have provided a detailed description of our analysis methodology and have also written the mathematical aspects of our analytical methods.

Comment 4. Overall, will it be possible to merge some of the sub-sections? Such as sample size and data management section may be discussed in data processing and analysis.

Response 4: We would like to thank you kindly for your comment. We had revised and corrected it accordingly

Comment5: Some sentences are not worded right?

Response 5: We would like to thank you kindly. We have revised and corrected it.

Results

Comment 1: Change in MDD consumption in Ethiopia- Line 196 – 197, I do not find table showing chi-square test. Will it be possible to include it? Same applies for line 203.

Response 1: We express our gratitude for your insightful comment. Certainly, we have included the relevant table in our report as Table 2.

Comment2: Line 218 – 219: Authors are claiming around 104.8% was due to the behavioral characteristics. Is this meaningful?

Response2: First and foremost, we express our sincere gratitude for your intriguing idea and thoughtful questions. We have gained valuable insights from them. Now, let’s delve into the specifics. The observed phenomenon arises from the inherent nature of multivariate decomposition analysis applied to a nonlinear (logit) model. Our objective is to examine trends and identify contributing factors related to changes in adequate minimum dietary diversity consumption among children. Specifically, we categorize this consumption (outcome variable) as either ‘adequate’ (coded as ‘1’) or ‘inadequate’ (coded as ‘0’). Each survey corresponds to a specific period: 2011 (labeled ‘0’) and 2016 (labeled ‘1’), as well as 2016 (labeled ‘0’) and 2019 (labeled ‘1’). Notably, in multivariate decomposition analysis for nonlinear (logit) models, percentage contributions can exceed 100%, contingent upon how the outcome variable is categorized. Importantly, this outcome is not indicative of an error; rather, it underscores the intricate interplay of characteristics and effects in explaining group differences. Even though certain behavioral factors remain unexplained in the model, this phenomenon remains meaningful. The two key reasons that account for percentage contribution exceeding 100% in our result could be due to nonlinear(logit) models of multivariate decomposition often dealing with nonlinear response models, where contributions do not linearly add up, leading to percentage contribution beyond 100%. Interaction effects: interaction terms between variables can amplify the impact of specific characteristics.

For example, mothers who were in the union are the only characteristic considered; the contribution due to union for women is 1251%, which exceeds 100%. This happens because the coefficient for the marital status of mothers who were in union was lower, leading to a larger relative contribution even though it is not statistically significant at a p-value less than 0.05 in our result.

We trust that you will comprehend our rationale, which is rooted in the inherent nature of the analysis and our specific interest in the outcome variable response coding.

Comment3: How did authors calculate percentage of contribution? Description needed.

Response 3: We sincerely appreciate your thoughtful comment. In our practical analysis, we utilized a regression model with Multivariate Decomposition Analysis for the logit model in STATA. To understand the percentage contribution in multivariate decomposition analysis, follow these steps:

Step 1: Calculate the standardized coefficients: These coefficients represent the effect of each predictor variable on the outcome variables.

Step 2: Take the absolute value and sum the absolute values: Next, sum up the absolute value of all the standardized coefficients. This total represents the combined effect of all predictor variables.

Step 3: Calculate individual contributions: Divide each standardized coefficient estimate by the sum obtained in Step 2. Multiply the result by 100 to express it as a percentage. This gives you the percentage contribution of each predictor variable to the outcome.

This method helps us understand how much each predictor contributes to the overall variation in the outcome. It’s a useful tool for analyzing the relative importance of different factors in explaining the observed differences or changes in a multivariate context.

Comment4: In table 2, I see some of the coefficients are zero. I am assuming the coefficients are close to zero. Since authors reported up to one digit after decimal, I am seeing it zero. The associated 95% confidence interval is also (0.0, 0.0). Such as for category rural of variable residence. But I am seeing the percentage of contribution is 14.7%. Will it be meaningful to refit this model with deleting variables with coefficients close to zero at least which are not significant? 

Response 4: We express our sincere gratitude for your insightful comment. As previously indicated in the table below as a note, the numbers were initially rounded to the nearest three decimal points, which do not precisely represent zero. To address any potential inconvenience, we have rectified the table by reporting the results rounded to the nearest five decimal points.

Comment5. Revising presentation on table 2.

Response 5: We express our sincere appreciation for your insightful comment. Following your feedback, we have made revisions and incorporated the changes into Table 3 of our updated version.

Comment6: applies on line 247. 

Response 6: We extend our sincere appreciation for your thoughtful comment. In response to your query, we have provided further clarification in response 2. We trust that you will comprehend our rationale, which is rooted in the inherent nature of the analysis and our specific interest in the outcome variable response coding.

Comment7: applies on table 3.

Response 7: We express our sincere appreciation for your insightful comment. Following your feedback, we have made revisions and incorporated the changes into Table 4 of our updated version.

Comment8: From table 3, I see, the percentage of increase due to the category union of marital status is 1251.45% and the percentage of reduction due to the category rural of variable residence is 151.160%. Second one is significant. Are they meaningful?

Response8: First and foremost, we express our sincere gratitude for your intriguing idea and thoughtful questions. In response to your query, we have provided further clarification in response 2. We have gained valuable insights from them. Now, let’s delve into the specifics. The observed phenomenon arises from the inherent nature of multivariate decomposition analysis applied to a nonlinear (logit) model. Notably, in multivariate decomposition analysis for nonlinear (logit) models, percentage contributions can exceed 100%, contingent upon how the outcome variable is categorized. Importantly, this outcome is not indicative of an error; rather, it underscores the intricate interplay of characteristics and effects in explaining group differences. Even though certain behavioral factors remain unexplained in the model, this phenomenon remains meaningful. The two key reasons that account for percentage contribution exceeding 100% in our model could be due to nonlinear(logit) models of multivariate decomposition often dealing with nonlinear response models, where contributions do not linearly add up, leading to percentage contribution beyond 100%. Interaction effects: interaction terms between variables can amplify the impact of specific characteristics. 

For children who were in rural residences the only behavioral characteristic considered; the contribution due to being from rural residence is 151%, which exceeds 100%. This happens because the coefficient for children who were from rural residences was lower, leading to a larger relative contribution which is statistically significant at a p-value less than 0.05 in our result.

Comment9: Will it be meaning to describe how difference due to characters and due to the effect of characters differ?

Response9: We would like to thank you kindly for your deep comment. Multivariate decomposition aims to dissect group differences in average predictions from models. It partitions the components of a group difference into a first change due to compositional differences (endowments): These arise due to variations in characteristics or endowments. Difference due to characters means that the change in MDD consumption was attributed by changes in population structure like age, marital status, educational status, ……..wealth index. For example: in our study, composition change in the household wealth index (percentage of households classified as middle and rich decreased from 44% to 16% and 42% to 17% from 2016 to 2019) contributed to the reduction of adequate MDD consumption by 7.39% and 2.95% respectively. “In 2019, it became evident that the population was economically worse off compared to 2016. Given that community wealth is closely tied to nutritional feeding practices, those who were economically disadvantaged experienced income shortages, which in turn impacted the feeding practices of their children.”

Secondly, changes resulting from specific characteristics impact various outcomes, such as returns, coefficients, or behavioral responses. As the composition of the population shifts, it influences behavioral responses either positively or negatively concerning certain practices. For instance, in our study, we observed that the behavior associated with higher age of mothers aged 35-49 contributed to a 45% increase in the consumption of adequate MDD (Minimum Dietary Diversity) from 2011 to 2016. This effect may be attributed to the accumulation of experiential knowledge (positive behavior) as mothers age increased, leading to improved feeding practices for their children.”

Reviewer #2: Thank you for the opportunity to review this study. The authors attempted to determine the trend of MDD and its determinants among children aged 6 to 23 months in Ethiopia from 2011-2019. While the study is good, I have some comments to strengthen the paper.

Response: We would like to thank you kindly for your deep comment. We revised and corrected it

Abstract

Comment1. Line 45: Specify the particular SDG target.

Response 1: We would like to thank you kindly for your deep comment. We revised and corrected it

Comment2. Line 53: Did you do a trend analysis or a change analysis?

Response 2: We sincerely appreciate your thoughtful comment. The issue you raised is indeed intriguing. Our study focused on trend analysis, specifically examining the trends in Minimum Dietary Diversity (MDD) consumption among children from 2011 to 2019 using nationally representative data. In this context, we used the term ‘change’ to highlight the determinants that contributed to either an increase or decrease in MDD consumption between 2011 and 2016, as well as between 2016 and 2019. To provide clarity, we have revised the title of our study to “Trend and Determinants of Minimum Dietary Diversity Consumption Change among Children Aged 6-23 months in Ethiopia from 2011 to 2019: A Multivariate Decomposition Analysis.” Our analysis delved into the trends of MDD consumption, revealing an increase from 2011 to 2016 followed by a decrease from 2016 to 2019. Through multivariate decomposition analysis, we explored the factors contributing to these changes in MDD consumption over the specified periods." 

Comment3. Line 61: Should be 'p-value' not 'P-value'

Response 3: We sincerely appreciate your thoughtful comment. We have carefully reviewed and made the necessary corrections.

Comment 4: Line 63: Change 'wasn't' to 'was not'.

Response 4: We sincerely appreciate your thoughtful comment. We have carefully reviewed and made the necessary corrections.

Comment5. The authors must rewrite the conclusion section. Lines 66-68 are just a repetition of the findings. At least, we should have a glimpse of the specific, practical recommendations in the conclusion.

Response 5: We sincerely appreciate your thoughtful comment. We have carefully reviewed and made the necessary corrections.

Background

Comment 6. The paragraphs in the background are too long which makes it difficult to get the main idea. Kindly restructure your paragraphs. A paragraph should contain one central theme.

Response 6: We sincerely appreciate your thoughtful comment. We have carefully reviewed and made the necessary corrections.

Comment 7. In the first paragraph, the authors present issues relating to dietary insufficiency. However, they fail to explain what this insufficiency is. There are no standards presented to show at what threshold a diet will be deemed insufficient. This should be presented.

Response 7: We sincerely appreciate your thoughtful comment. After careful review, we have adopted the standard definition established by The World Health Organization (WHO) and the United Nations Children’s Fund (UNICEF). According to this definition, Adequate Minimum Dietary Diversity (MDD) refers to a child who consumes at least five out of eight specified food groups. These food groups include breast milk, grains, roots, tubers, legumes, nuts, dairy products, flesh foods (such as meat, fish, poultry, and organ meats), eggs, vitamin-A-rich fruits, and other fruits and vegetables.

Comment8. I suggest a complete overhaul of t

---

## [Decision Letter · Decision Letter 1]

27 May 2024

PONE-D-24-02192R1Trend and Determinants of Minimum Dietary Diversity Consumption Change among Children Aged 6-23 months in Ethiopia From 2011 to 2019:  A Multivariate Decomposition AnalysisPLOS ONE

Dear Dr. Endawkie,

Thank you for submitting your manuscript to PLOS ONE. After careful consideration, we feel that it has merit but does not fully meet PLOS ONE’s publication criteria as it currently stands. Therefore, we invite you to submit a revised version of the manuscript that addresses the points raised during the review process.

We look forward to receiving your revised manuscript.

Kind regards,

Satyajit Kundu

Academic Editor

PLOS ONE

**Additional Editor Comments:**

Authors are requested to address the 2nd reviewer's comments carefully so that any further delay can be avoided to make a decision.

Reviewers' comments:

Reviewer's Responses to Questions

**Comments to the Author**

1. If the authors have adequately addressed your comments raised in a previous round of review and you feel that this manuscript is now acceptable for publication, you may indicate that here to bypass the “Comments to the Author” section, enter your conflict of interest statement in the “Confidential to Editor” section, and submit your "Accept" recommendation.

Reviewer #2: (No Response)

2. Is the manuscript technically sound, and do the data support the conclusions?

Reviewer #2: Partly

3. Has the statistical analysis been performed appropriately and rigorously? 

Reviewer #2: Yes

4. Have the authors made all data underlying the findings in their manuscript fully available?

Reviewer #2: Yes

5. Is the manuscript presented in an intelligible fashion and written in standard English?

Reviewer #2: Yes

6. Review Comments to the Author

**Reviewer #2: Dear authors**

**Thank you for responding to the comments raised. However, you did not respond to whether to accounted for collinearity. Also, I fail to see the theoretical perspective guiding this research. Why so?**

7. PLOS authors have the option to publish the peer review history of their article (what does this mean?). If published, this will include your full peer review and any attached files.

Reviewer #2: No

---

## [Author Response · Author response to Decision Letter 1]

14 Jun 2024

Authors’ Responses to Editor and Reviewers Comments

Dear Academic Editor, Plose One 

 “Trend and Determinants of Minimum Dietary Diversity Consumption Change among Children Aged 6-23 months in Ethiopia From 2011 to 2019: A Multivariate Decomposition Analysis”

Dear Plos One Editors and Reviewers;

We are thankful for your constructive comments. We have looked at the comments and have revised our paper accordingly. We hope our paper improved as a result of incorporating the reviewer’s and academic editor’s comments and suggestions. Here are the authors’ responses to the comments.

Please find for your kind consideration the following:

1. A revised manuscript with track changes.

2. A revised paper without tracked changes

3. A rebuttal letter that responds to each point raised by the academic editor and reviewer. 

The point-by-point responses of authors are written by hoping these changes would meet with your favorable consideration, we are happy to hear if there are more comments and suggestions. Please do not hesitate to let us know if you have any questions.

Yours Sincerely 

Mr. Abel Endawkie Correspondence Author 

Department of Epidemiology and Biostatistics School of Public Health College of Medicine and Health Science Wollo University Dessie Ethiopia 

Tel. 251935459310 Email address abelendawkie@gmail.com

We have tried our best to improve it accordingly:

 Please revise the manuscript.

https://journals.plos.org/plosone/s/file?id=wjVg/PLOSOne_formatting_sample_main_body.pdf andhttps://journals.plos.org/plosone/s/file?id=ba62/PLOSOne_formatting_sample_title_authors_affiliations.pdf

Response 1: We revised our manuscript based on Plos one submission guidelines (manuscript, title format, and table format). 

Point-by-point response of Authors for editors' and reviewers' comment 

Editor's Comments to the Authors

Additional Editor Comments:

The authors are requested to address the reviewers' comments carefully and in detail 

Response: we are grateful to thank you for your kindness and response. We tried to correct it thoroughly. 

Reviewer Comment to the Authors 

Reviewer#2 comment: Thank you for responding to the comments raised. However, you did not respond to whether to accounted for collinearity. Also, I fail to see the theoretical perspective guiding this research. Why so?

Response: Thank you for your feedback and for taking the time to review the comments. I apologize for any inconvenience caused by my previous response. Regarding the issue of collinearity, I would like to assure you that it has been duly accounted for in our research. In terms of the theoretical perspective guiding our research, we have indeed considered this aspect. The theoretical framework of our study is grounded in an insert theoretical framework, which provides a foundation for understanding the phenomena under investigation. We have thoroughly reviewed the relevant literature and integrated existing theories to inform our research design and analysis.

The response is provided below in detail.

Comment 1: Did you account for collinearity? 

Response 1: First and foremost, we express our sincere gratitude for your intriguing idea and thoughtful questions. We would like to sorry for this inconvenience. The inconvenience happened due to we did not report the variable selection process and multicollinearity assessment as we believe it was long even it affected our transparency and reproducibility for the scientific work. We value your insights and suggestion since Multicollinearity is a problem when independent variables are correlated and analyzed in the model.

As we know Multicollinearity occurs when multiple independent variables in a regression model are highly correlated with each other and Multivariate decomposition analysis, aims to partition differences in mean responses between groups or over time into components that reflect both the differences in mean levels of model predictors and the differences in the effects of those predictors across groups or periods. 

Even though these two concepts address different aspects of statistical analysis, they can be combined to understand the impact of multicollinearity on the decomposition of group differences. The way to detect multicollinearity in multivariate decomposition differs from the usual work of other regression models. 

Detecting Multicollinearity:

The most common way to detect multicollinearity is by using the variance inflation factor (VIF). VIF measures the correlation and strength of correlation between predictor variables in a linear regression model. High VIF values (typically above 5 or 10) indicate strong multicollinearity

Multicollinearity was checked among explanatory variables by using average standard error at a cut of point±2 for the logistic model.

Because Multicollinearity detection in multivariate decomposition is before performing multivariate decomposition, we simply screened the data whether it has Multicollinearity or not using the logistic model before we conducted multivariate decomposition because the decomposition is multivariate decomposition for the logit model. Not only this the variable selection was conducted before multivariate decomposition analysis using Bivariable logistic regression of p-value <0.25.

In our analysis, we assessed Multicollinearity using the average standard error of each standard error since we had a binary outcome variable and we used logistic regression. 

Since it is important we shortly wrote the variable selection process and multicollinearity assessment to develop a multivariate decomposition analysis in the method section as below.

Before analysis first, bi-variable logistic regression was fitted and variables with p-values less than 0.25 were selected to develop multivariate decomposition analysis. Multicollinearity was checked among explanatory variables by using average standard error at a cut of point±2 and there is no Multicollinearity since the standard errors were between±2.

The observed procedure arises due to the inherent nature of multivariate decomposition analysis applied to a nonlinear (logit) model.

 Comment 2: Given the nature of this study, it will be advisable for the authors to include a theoretical framework to strengthen their findings and lay the foundation for their arguments in the discussion section 

Response 2: First and foremost, we express our sincere gratitude for your intriguing ideas and thoughtful questions. We would like to sorry for this inconvenience. As you suggest the theoretical framework helps to strengthen the findings and discussion:

In the context of child feeding practices and MDD consumption, the framework suggests that the trend of MDD consumption change is determined by the change in different factors: including composition change at the individual level like child characteristics (age, sex), maternal characteristics (age, education, occupation), composition change at the household characteristics (wealth status, household head sex, number of under-five living children, and household size), composition change at the community-level factors: residence, region, and behavioral change (coefficient) (figure 1). Then the trends and determinants of MDD consumption change found in this study can be interpreted through the lens of this theoretical framework that changes in individual, household, and community-level factors over time likely contributed to the observed trends in MDD consumption change. The conceptual framework was developed using literature.

The figure is provided next to the idea of the paragraph based on the journal's guideline 

 We appreciate your kind response and genuine communication. We hope these changes will meet your favorable consideration and we are happy to hear if there are more comments and suggestions. Please do not hesitate to let us know if you have any questions.

We feel all of you may understand our feelings and interests.

We are thankful for all!!

---

## [Decision Letter · Decision Letter 2]

23 Jul 2024

Trend and Determinants of Minimum Dietary Diversity Consumption Change among Children Aged 6-23 months in Ethiopia From 2011 to 2019:  A Multivariate Decomposition Analysis

PONE-D-24-02192R2

Dear Dr. Abel Endawkie,

We’re pleased to inform you that your manuscript has been judged scientifically suitable for publication and will be formally accepted for publication once it meets all outstanding technical requirements.

Kind regards,

Satyajit Kundu

Academic Editor

PLOS ONE

Additional Editor Comments (optional):

After closely examining your revised version, we are pleased to notify you that the current version meets the requirements for publication in this journal.

Reviewers' comments:

Reviewer's Responses to Questions

**Comments to the Author**

1. If the authors have adequately addressed your comments raised in a previous round of review and you feel that this manuscript is now acceptable for publication, you may indicate that here to bypass the “Comments to the Author” section, enter your conflict of interest statement in the “Confidential to Editor” section, and submit your "Accept" recommendation.

Reviewer #2: All comments have been addressed

2. Is the manuscript technically sound, and do the data support the conclusions?

Reviewer #2: Yes

3. Has the statistical analysis been performed appropriately and rigorously? 

Reviewer #2: Yes

4. Have the authors made all data underlying the findings in their manuscript fully available?

Reviewer #2: Yes

5. Is the manuscript presented in an intelligible fashion and written in standard English?

Reviewer #2: Yes

6. Review Comments to the Author

Reviewer #2: I commend the authors for responding to all comments raised. I support the publication of this paper.

7. PLOS authors have the option to publish the peer review history of their article (what does this mean?). If published, this will include your full peer review and any attached files.

Reviewer #2: No

---

## [Editor Report · Acceptance letter]

25 Jul 2024

PONE-D-24-02192R2 

PLOS ONE

Dear Dr. Endawkie, 

I'm pleased to inform you that your manuscript has been deemed suitable for publication in PLOS ONE. Congratulations! Your manuscript is now being handed over to our production team.

Kind regards, 

on behalf of

Dr. Satyajit Kundu 

Academic Editor

PLOS ONE